# Influences on Teachers' Intention to Apply Classroom Management Strategies for Students with ADHD: A Model Analysis

**Anna Enrica Strelow** [1],*, **Martina Dort** [1] , **Malte Schwinger** [2] **and Hanna Christiansen** [1]

1 Clinical Child and Adolescent Psychology, Department of Psychology, University of Marburg, 35032 Marburg, Germany; martina.dort@uni-marburg.de (M.D.); hanna.christiansen@uni-marburg.de (H.C.)
2 Educational Psychology, Department of Psychology, University of Marburg, 35032 Marburg, Germany; malte.schwinger@uni-marburg.de
* Correspondence: anna.strelow@uni-marburg.de

**Abstract:** Students with attention deficit/hyperactivity disorder (ADHD) show reduced on-task behavior at school and educational problems due to the symptoms associated with this diagnosis. Classroom management strategies (CMS) are important to reduce impairment due to ADHD symptoms but are not yet well implemented. In this study we analyzed whether the facilitators and barriers regarding the intention to apply CMS identified for pre-service teachers are replicable in a sample of teachers in service. Overall, 599 teachers in service completed an online survey on the intention to apply CMS, their attitude towards CMS and towards students with ADHD, direct experiences, individual differences, and social influences. We calculated path models that significantly clarified variance in the intention to apply CMS ($R^2_{\text{intention to use effective CMS}}$ = 0.47, $p < 0.01$ and $R^2_{\text{intention to use ineffective CMS}}$ = 0.39, $p < 0.01$). It turns out that similar variables are relevant to teachers in service as well as pre-service teachers. A models' extension to include variables that do justice to the difference between the two groups, such as work experience, shows a better model fit. Especially, attitude towards CMS, attitude towards students with ADHD, strain, perceived behavioral control and teachers' affiliation with primary or special needs schools are important variables regarding the intention to apply CMS. The implementation of effective and elimination of ineffective CMS should thus be addressed by targeting teacher's attitudes towards children with ADHD. Furthermore, strain prevention and education might enhance the application of effective CMS.

**Keywords:** ADHD; attitude; classroom; classroom management strategies; expectation; intervention; knowledge; teachers





## 1. Introduction

### 1.1. Attention Deficit/Hyperactivity Disorder and Effective Treatment

To make a diagnosis of attention deficit/hyperactivity disorder (ADHD), the core symptoms inattention, hyperactivity, and impulsivity need to manifest cross-situationally, e.g., at home and at school, and result in clinical impairment [1,2]. ADHD affects five to seven percent of students in schools [3–6]. Symptoms like failing to pay attention to details, having difficulty sustaining attention or organizing a task, or excessive talking or fidgeting cause reduced on-task behavior at school and often result in educational problems that are predictive for underachievement and a heightened risk for delinquency [7–11].

Pharmacological treatment that targets those core symptoms is the most frequent intervention for students with ADHD, and combining medication and behavioral therapy (as well as each on its own) reduce ADHD symptoms, but the effects are attenuated with respect to school achievement [12,13]. As the impairment is specifically serious at school, students are often referred for diagnostics and treatment after entering school.

This highlights the need for treatments specifically targeting impairments associated with schools.

Evidence-based classroom management strategies (CMS) are known to effectively reduce disruptive classroom behavior of students with ADHD [14]. Therefore, CMS are an important, impactful treatment module that cannot just reduce the specific academic impairments, they might also influence the long-term negative effects associated with ADHD such as lower educational accomplishments [14,15]. Finally, this might enhance the chance for equal academic achievement for all students. Thus, using CMS in the classroom leads to a sustainable and long-term change in behavior among the affected students, whereas with other treatment options, such as medication, symptoms often return after termination of therapy. Furthermore, the use of CMS is not only effective for students with ADHD [14]. Teachers can successfully use the strategies they have learned with other students resulting in gain of resources in the long term and overall relieving the burden on teachers.

A meta-analysis [14] supported previous studies' findings that demonstrate the effectiveness of such CMS in reducing unfavorable classroom behavior. Nevertheless, teachers rely more often on ineffective (e.g., threatening the student with punishment) than effective CMS [16,17]. Due to this gap between science and practice, a lot of potential is lost on part of both the teachers and students [17,18].

### 1.2. Reasons for the Observed Science-Practitioner Gap

One reason for this gap between science and practice is the reduced communication between the scientific fields of psychology/psychiatry and education [19]. Most research comes from the field of psychology/psychiatry and fails to take the teachers' perspective [19]. This might be why effective CMS have not found their way into teachers' daily school routines yet.

When asking for contributions to facilitating effective CMS implementation, an initial model analysis with data from 1086 pre-service teachers revealed that the behavioral attitude towards CMS is the most important variable in explaining variance in the intention to apply effective and ineffective CMS [20]. According to Ajzen [21], the attitude towards a behavior (e.g., applying a CMS) is the expectation that the behavior will have (positive) consequences. This implies that the expectation that the CMS will induce a favorable outcome is the most important predictor for the intention to apply it.

An Austrian study that included teachers in service and teachers not regularly working in schools showed also that behavioral attitude is the most important variable in clarifying variance in the intention to apply CMS [22]. Furthermore, their study demonstrated significant differences between in-service and pre-service teachers [22].

According to Theory of Planned Behavior (TPB) by Ajzen [21], there are two components of attitude. Next to the behavioral attitude, he defines the attitude towards an object as a further variable that in turn influences one's behavioral attitude [21]. In our context, the attitude towards an object reflects the attitude of teachers towards affected students. In our study with pre-service teachers, we demonstrated that the attitude towards students with ADHD also impacted the intention to apply CMS [20]. Moreover, other variables such as direct experiences, social influences, and individual differences influence attitudes and behavior intention according to the violation-expectation (VoilEx) model by influencing expectations that represent the behavioral attitude [20,23].

### 1.3. Expectations Defined as Behavioral Attitude

The VioilEx model defines expectations as relations between specific situations (applying CMS for a student) and associated responses (CMS fails to work, and student continues behaving disruptively) resulting from learning processes. The exemplarily described learning process results in a negative behavioral attitude towards the CMS applied that in turn lowers the intention to apply it [20,23]. These expectations and therefore behavioral attitude are influenced by specific experiences, social influences, and individual differences [20,23].

We further discuss these aspects in the sections below. Figure 1, taken from Strelow, Dort, Schwinger, and Christiansen, illustrates expectations' influences and impacts [20].

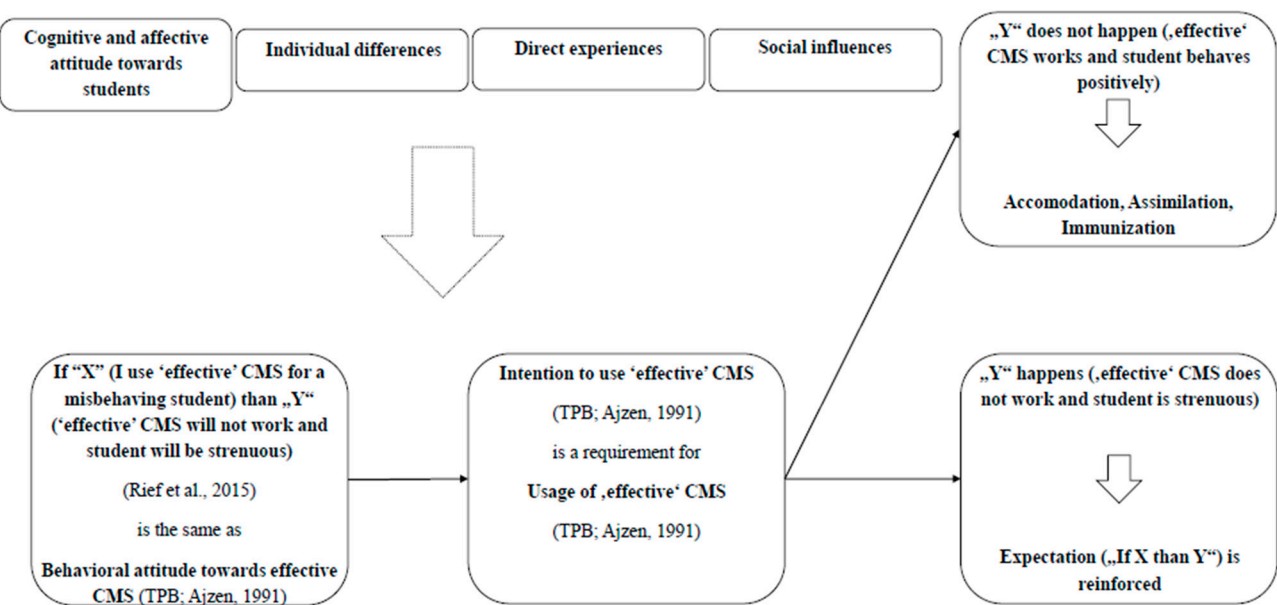

**Figure 1.** We define expectations as the behavioral attitude. Cognitive and affective attitude, individual differences, direct experiences, and social influences influence expectations and therefore the behavioral attitude. The behavioral attitude influences the intention and applying of classroom management strategies. Depending on whether the expectation is unfulfilled or confirmed, there will be a learning, reinforcement, or no influence on the expectation. Figure is taken from Strelow, Dort, Christiansen, and Schwinger [20] with kind approval.

### 1.4. Influences on Attitude and the Intention to Apply CMS

#### 1.4.1. Direct Experiences

Direct experiences with students affected by ADHD led teachers to classify them as stressful [24], and this self-perceived individual stress correlated positively with the intention of pre-service teachers to apply ineffective CMS [20]. Further, more direct experiences with students with ADHD result in stronger negative responses towards them [24]. An investigation that focused on the differences between pre- and in-service teachers showed that teachers with more direct experiences responded more negatively towards students with ADHD [24]. Most likely this is associated with work experience, e.g. more contact with such students. Work experience as assessed by numbers of years on the job is thus a variable that differentiates between pre- and in-service teachers and between teachers in service. Therefore, work experience may be an additional potential facilitator of effective or ineffective CMS.

Apart from this, experience differs according to the education of teachers, i.e., whether they have been trained to teach students with special needs, or those in primary or secondary school. The majority of research on CMS or ADHD focuses on primary school teachers [25–27]. Additionally, there is a difference in knowledge between teachers who instruct students with special needs (and were trained to do so) and those who do not [28]. In contrast to main, middle, or academic high school teachers, primary and special needs school teachers are trained to support a wide range of students and might therefore be given other information and other strategies during their training. The work experience assessed by years on the job and the education received are therefore important variables when assessing potential influences on the intention to apply CMS.

### 1.4.2. Social Influences

A social influence is a subjective norm, namely (according to Ajzen) the willingness to behave according to the expectations of relevant peers (e.g., "I implement effective CMS in my classroom, as my school values those") [21,29]. The subjective norm is perceived as a relevant factor for forming attitudes and behavioral intentions [21,29], though it did not prove to be a significant predictor in our study on pre-service teachers' attitudes towards children with ADHD and their intention to implement CMS [20]. As pre-service teachers do not regularly teach in schools, their social environment might differ from that of in-service teachers. In-service teachers might differ in this respect also.

### 1.4.3. Individual Differences

Individual differences also influence how attitudes and behavioral intentions are formed. Personality factors like the big five (openness, conscientiousness, extraversion, agreeableness, and neuroticism) as well as social dominance orientation (SDO) and right-wing authoritarianism (RWA) are relevant variables in this context, and influence the intention of pre-service teachers to apply CMS [30]. Another individual difference is psychological strain. Previous research also shows that teachers often experience psychological strain, which is associated with their attitude towards students with ADHD [31,32]. More precisely, psychological strain is linked to a weaker intention in pre-service teachers to implement effective CMS [20]. However, pre-service teachers might not feel the same psychological strain as in-service teachers. To get an impression of how much strain affects in-service teachers, this needs closer examination. Stress reactivity is a variable revealing the intensity of individual reactions to stressful situations, which are known to differ individually among people [33]. It describes why some people react differently to stress in the same situation [33]. People who score low on stress reactivity are able to handle stressful events in a calm and confident manner, and can set themselves appropriate goals [33]. This is linked to a stronger intention in pre-service teachers to apply effective CMS, and is another relevant individual difference [20].

Knowledge as another individual difference in conjunction with ADHD proved to affect the intention of pre-service teachers to implement CMS [20]. Pre-service teachers might differ in knowledge from those in service [20,34]. These differences can result from the fact that the training teachers receive may have improved in recent years, and more information about ADHD might already be included in university teacher-training. On the other hand, in-service teachers have had contact with many students, some of whom might have been affected by ADHD. Thus, in-service teachers might already have profound knowledge about ADHD. Perceived behavioral control is also an individual difference, and with respect to the TPB an important predictor of behavior [29]. It is defined as the self-assessed expectation that one can execute a behavior. In CMS terms, this would be the conviction that a teacher is able and has all means available to carry them out [21,29]. In our investigation with pre-service teachers, perceived behavioral control correlated negatively with the intention to apply ineffective CMS [20]. It is therefore important to examine whether this also applies to in-service teachers.

Individual demographic differences such as sex have a direct effect on teaching behavior. There is evidence that male teachers teach differently than female teachers [35]. However, to our knowledge, this effect has not yet been verified in terms of classroom strategies. Nevertheless, this variable might clarify variance in the intention to apply CMS. Furthermore, as work experience and age correlate closely, it is not entirely clear whether work experience's effect might not also be explained by age. We therefore examined sex and age in this investigation.

### 1.5. The Present Study

Summing up, previous research indicates that attitude shapes the intention to apply CMS. Attitudes are influenced by direct experiences, social influences, and individual differences. As outlined above, pre- and in-service teachers differ in some variables.

Therefore, the present study examines whether the model for pre-service teachers can be replicated in a sample of in-service teachers and whether expanding upon the model with the variables sex, age, and work experience (according to years on the job) and type of school (primary and special needs school vs. secondary school) improves the model fit. Our study therefore posed these questions:

(1) Can the model to implement (in-)effective CMS for pre-service teachers (with the variables attitudes, direct experiences, social influences, individual differences) be replicated in a sample of in-service teachers?

(2) Does expanding the model for in-service teachers by considering the variables sex, age, and work experience according to years on the job and type of school result in an improved model fit?

## 2. Materials and Methods

### 2.1. Sample Population

The online tool SoSciSurvey (https://www.soscisurvey.de/) was used in the present study with teachers from Germany. The survey was started by $N = 2320$ participants. The final sample population included $N = 635$ participants who completed the survey (drop-out rate 72.63%). Overall, 5.67% ($n = 36$) of the participants who completed the survey were excluded from the analysis, because they completed less than 75% of at least one scale, thus our analysis was based on $N = 599$ teachers. The mean age of the participants was $M = 41.33$ years ($SD = 10.01$). The participants subdivided into 17.7% males, and 82.3% females; no one identified as diverse. Details on the sample and the type of school participants taught at are illustrated in Table 1.

**Table 1.** The school where teachers claimed to teach. Information about N = 599 participants.

| School Type | % | Number (Absolute) |
|---|---|---|
| Primary school (Grundschule) | 54.3 | 325 |
| Comprehensive school (Gesamtschule) | 20.0 | 120 |
| Main school (Hauptschule) | 7.5 | 45 |
| Middle school (Realschule) | 8.3 | 50 |
| Academic high school (Gymnasium) | 7.2 | 43 |
| Special needs school (Förderschule) | 9.3 | 56 |
| Other (Sonstige) | 5.5 | 33 |

Note: The percentages can exceed 100. Due to the school systems in Germany, more than one school could be selected.

The majority of teachers were teaching in Hessia (59.6%) and Lower Saxony (29.9%). Only a few teachers from other federal states participated in this study, although every federal state contributed participants. These ranged between 3.8% (Saarland) and 0.2% (Berlin, Brandenburg, Bremen, Mecklenburg-Western Pomerania). The teachers stated that they had $M = 13.27$ ($SD = 9.14$) years of work experience with a minimum of one year and a maximum of 42 years. In addition, the teachers stated that during their career they instructed $M = 16.27$ ($SD = 24.24$) students with ADHD with a minimum of one and a maximum of 150 students.

### 2.2. Procedure and Sequence of the Survey

E-mail addresses of all school types were compiled in all federal states via the official schools' internet pages. Afterwards, we sent the link leading to the survey via e-mail to schools with the request that it be passed on to teachers.

The link was also distributed via Facebook groups to teachers. The subject lines in the e-mail and Facebook posts included information on possible prizes for participation, to be awarded via a raffle. It was possible to win a Nintendo Switch, a voucher for a wellness-weekend, or two tickets for a musical (approximate value 300 Euros). The e-mails and Facebook posts also included a cover note that informed about the study, the raffle,

and the link to the survey. The first page of the survey included detailed information on the purpose of the study, data collection and storage, and informed about the possibility to drop out of the study at any time with no consequences. All participants were asked to comply with those requirements, otherwise, they could not proceed with the survey. Apart from the sociodemographic information, our survey included questions on variables that had been identified as relevant for predicting pre-service teacher classroom behavior [20]. Those included questions on personality, knowledge, attitudes towards CMS and students with ADHD, and strategies to handle students' classroom behaviors (see measure section for details). At the start of the survey, a short case vignette depicting a typical student with ADHD was presented as a priming stimulus. The survey ended with the option to receive further information on the study and an option to participate in the raffle. The survey was online between April and October 2018.

### 2.3. Measures

As this study is a replication of our study that incorporated the same survey completed by pre-service teachers [20], only a few adjustments were made with respect to wording to fit the in-service sample; some scales were shortened based on prior results.

#### 2.3.1. Intention to Implement Classroom Management Strategies (CMS)

The ADHD School Expectation Questionnaire (ASE) was used to assess CMS of teachers when handling ADHD-related behavior in their classrooms [36]. The ASE differentiates between effective and ineffective strategies using a 12-point Visual Analogue Scale (VAS) ranging from never to very often. Values are transformed into an interval scale from zero to one. The subscale on effective strategies has 15 items, the one on ineffective ones 12. Psychometric properties are satisfactory with Cronbach's $\alpha = 0.81$ for the effective CMS scale ($M = 0.74$, $SD = 0.09$), and $\alpha = 0.74$ ($M = 0.36$, SD = 0.14) for the ineffective one; total scale: $\alpha = 0.72$ ($M = 0.57$, $SD = 0.09$).

#### 2.3.2. Attitude towards Students with ADHD and towards Classroom Management Strategies (CMS)

The ASE further assesses affective and cognitive attitudes of teachers towards their students with 33 items on a 12-point VAS [36]. Items measuring the cognitive part describe different behavior patterns, and the rater indicates whether that behavior is likely or unlikely (coded from 0 to 1) in a student with ADHD and whether that behavior is perceived as positive or negative (coded from −3 to +3). In reference to Ajzen [29], the behavior likelihood and associated positive or negative ratings are multiplied afterwards. The affective part of the scale includes items that assess the emotions a teacher may feel towards students with ADHD. Those are rated as likely or unlikely (coded from 0 to 1) and whether they are negative or positive (coded from −3 to +3). Internal consistency is satisfactory with $\alpha = 0.85$ ($M = -13.12$, $SD = 14.66$).

The behavioral attitude subscale consists of 28 items that are rated on a 12-point VAS ranging between not effective at all and very effective. With reference to Ajzen [21,29] the behavioral attitude is assessed on the basis of the behaviors' expected effectiveness. Values are re-coded from zero to one. This scale consists of the two subscales attitude towards the effective CMS ($\alpha = 0.87$, $M = 0.76$, $SD = 0.14$) and the attitude towards ineffective CMS ($\alpha = 0.71$, $M = 0.26$, $SD = 0.13$). Cronbach's alpha for the total scale is $\alpha = 0.75$ ($M = 0.54$, $SD = 0.09$).

#### 2.3.3. Direct Experiences
Individual Stress through ADHD in the Classroom

Teachers' individual stress levels are assessed with one item ("How high would you rate your stress due to the behavior of students with ADHD in the classroom?"). This item is also scored on a 12-point VAS ranging from not at all to very massive and transformed afterwards to range from zero to five. The mean of the item is M = 3.28, SD = 1.25.

Work Experience Assessed in Years and School Type at Which They Were Trained to Teach

We investigated work experience as years the participants had taught as a teacher ($M = 13.27$, $SD = 9.14$, $R = 1_{min}$–$42_{max}$). Furthermore, they informed us about the type of school where they teach (further results are depicted in Section 2.1). For this investigation, we put the primary and special group teachers in one group, and the others in a comparison group.

### 2.3.4. Social Influences
Subjective Norm

Subjective norm is assessed with six items based on the TPB. A 12-point VAS (I totally disagree to I totally agree) is used that later ranges from zero to five. This scale revealed unacceptable Cronbach's alpha values ($\alpha = 0.15$, $M = 2.95$, $SD = 0.61$). A shortened version with three items leads to better, but still unsatisfactory values: $\alpha = 0.51$ ($M = 3.27$, $SD = 1.00$); discriminatory $r_{lt} = 0.29$ to $r_{lt} = 0.41$. We further used the subjective norms' short version in the presents study.

### 2.3.5. Individual Differences
Big Five Personality Traits

To assess the big five personality factors, we employed the German short version of the Big Five Inventory (BFI-K) [37,38] with a five-point Likert scale (very inaccurate to very applicable). The psychometric properties were satisfactory in our study with extraversion $\alpha = 0.75$ ($M = 3.89$; $SD = 0.65$), conscientiousness $\alpha = 0.71$ ($M = 3.98$; $SD = 0.61$), neuroticism $\alpha = 0.66$ ($M = 2.60$; $SD = 0.67$), and openness $\alpha = 0.75$ ($M = 4.00$; $SD = 0.65$). The value for agreeableness $\alpha = 0.52$ ($M = 3.73$; $SD = 0.52$) was not satisfactory.

Right-Wing Authoritarianism (RWA)

RWA was assessed via the German short version on right-wing authoritarianism (KSA-3) by [39]. RWA's aggregated value is measured using a six-point Likert-type scale (I totally disagree to I totally agree). The overall Cronbach's $\alpha$ is 0.62 ($M = 2.09$; $SD = 0.94$).

Social Dominance Orientation (SDO)

The 12-item German SDO scale ([40] measures SDO on a six-point Likert-type scale (I totally disagree to I totally agree). Cronbach's $\alpha = 0.80$ ($M = 1.16$, $SD = 0.65$) in the present study.

Psychological Strain

Psychological strain is assessed via the short version of the Brief Symptom Inventory (BSI) [41,42] that consists of 18 items rated on a four-point Likert scale ranging from nothing at all to very strong. The Global Severity Index (GSI) summarizes all subscales and is therefore used in this survey as a marker of psychological strain. Internal consistency is satisfactory for this study with $\alpha = 0.83$ ($M = 3.57$, $SD = 0.63$) for the GSI.

Stress Reactivity

The 23-item Perceived Stress Reactivity Scale (PSRS) with five subscales and one total scale was used [43] to measure stress reactivity (e.g., When I want to relax after a hard day at work … ) with these possible answers: "That's usually quite difficult for me, I usually succeed, I usually have no problem at all." The total score used in this study has an acceptable Cronbach's $\alpha$ value of 0.88 ($M = 20.79$, $SD = 7.46$).

Knowledge

Knowledge was measured with the corresponding ASE scale including 24 items [36]. Correct answers are credited with 1; incorrect ones with 0. Altogether, 24 credits can be obtained. Cronbach's $\alpha$ is satisfactory with 0.78 ($M = 9.25$, $SD = 4.29$).

Perceived Behavioral Control

Two items based on the TPB were generated to measure perceived behavioral control. The answers were assessed with a 12-point VAS (I totally disagree to I totally agree) and were transformed afterwards in a scale from zero to five. Cronbach's alpha was $\alpha = 0.79$ ($M = 2.82$, $SD = 1.24$) and showed a discriminatory power of $r_{lt} = 0.66$.

*2.4. Statistical Analyses*

All analyses were conducted using IBM® SPSS®24.0 and Mplus 8.4 [44]. All raw data are stored at the Department of Clinical Child and Adolescent Psychology at Philipps University Marburg, Germany. At first, inter-correlations between the assessed variables are shown ($|r| = 0.10$ is defined as a low, $|r| = 0.30$ as a moderate, and $|r| = 0.50$ as strong coherence [45]). Linear path analysis with Mplus 8.4 [44] is used to display, calculate, and check complex relationships between variables, and to identify the explained variance. Mediation analyses are carried out using bootstrapping with 10,000 iterations examining significant indirect effects. Standardized path coefficients were calculated. As fit indices squared-chi-tests, the Comparative-Fit-Index (*CFI*; should be higher than 0.95, best above 0.97), the Tucker–Lewis-Index (*TLI*; should exceed 0.95, best above 0.97), Root-Mean-Square-Error-of-Approximation (*RMSEA*; should be under 0.05), and Standardized-Root-Mean-Square-Residual (*SRMR*; should be under 0.05) were used [46]. Additionally, $R^2$ was used to estimate how high the explained variance of the variables is. A value of 0.26 is rated as high variance explanation, a value of 0.13 as moderate and a value of 0.02 as low [45].

First, the path analyses that fitted best for pre-service teachers [20] were replicated. Second, the following variables were added: sex, age, primary school/special needs school teachers, work experience assessed in years and school type they were educated to teach at.

## 3. Results

*3.1. First Research Question ("Can the Model to Apply (In-)Effective CMS for Pre-Service Teachers Be Replicated in a Sample of In-Service Teachers?")*

To answer our first research question, we replicated the variables that fit best for pre-service teachers [20]. This led to an unsatisfactory fit regarding the effective CMS' model ($R^2_{intention} = 0.37^{**}$; $R^2_{behav.\ attitude} = 0.17^{**}$; test of model fit: $\chi^2(3) = 36.56$, $p < 0.01$; test for the baseline model: $\chi^2(29) = 394.00$, $p < 0.01$; *CFI* = 0.91; *TLI* = 0.11; *RMSEA* = 0.15; *SRMR* = 0.04).

The replication investigation for ineffective CMS resulted in a good model fit ($R^2_{intention} = 0.36^{**}$; $R^2_{behav.\ attitude} = 0.09^{**}$; test of model fit: $\chi^2(8) = 6.01$, $p = 0.65$; test for the baseline model: $\chi^2(29) = 284.370$, $p < 0.01$; *CFI* = 1.00; *TLI* = 1.00; *RMSEA* = 0.09; *SRMR* = 0.02).

*3.2. Second Research Question ("Does Expanding the Model for In-Service Teachers Result in an Improved Model Fit?")*

To answer our second research question, we conducted two further analyses with the above-mentioned added variables (sex, age, primary school/special needs school teachers, work experience assessed in years and school type they were educated to teach at).

3.2.1. Effective Classroom Management Strategies

Our fit indices reveal a good model fit for the model regarding the intention to apply effective CMS with the additional variables (test of model fit: $\chi^2(2) = 1.13$, $p = 0.57$; test for the baseline model: $\chi^2(39) = 458.81$, $p < 0.01$; *CFI* = 1.00; *TLI* = 1.00; *RMSEA* = 0.00; *SRMR* = 0.01) and a high variance explanation ($R^2_{intention} = 0.47$, $p < 0.01$; $R^2_{behav.\ attitude} = 0.24$, $p < 0.01$). The model is depicted in Figure 2. For the intention to implement effective CMS overall 47% ($R^2 = 0.47$) of the variance were explained, equaling high clarification as well [45]. Variables positively contributing were behavioral attitude towards effective CMS ($\beta = 0.42$, $p < 0.01$), perceived behavioral control ($\beta = 0.18$, $p < 0.01$), conscientiousness ($\beta = 0.11$, $p < 0.01$), individual stress through ADHD ($\beta = 0.14$, $p < 0.01$), teachers affiliated

with a primary or special needs school ($\beta = 0.19$, $p < 0.01$). Stress reactivity ($\beta = -0.15$, $p < 0.01$) revealed a negative impact. The variance clarification regarding the behavioral attitude towards effective CMS was at 24% ($R^2 = 0.24$). Variables with a positive influence were knowledge ($\beta = 0.21$, $p < 0.01$), sex ($\beta = 0.13$, $p < 0.01$), perceived behavioral control ($\beta = 0.21$, $p < 0.01$), extraversion ($\beta = 0.12$, $p < 0.01$), attitude towards students with ADHD ($\beta = 0.14$, $p < 0.01$), and teachers affiliated with a primary or special needs school ($\beta = 0.09$, $p < 0.05$). All mediations with the intention to apply effective CMS as the dependent and the behavioral attitude towards CMS as the mediating variable are significant: knowledge ($\beta = 0.09$; 95%-CI: 0.06, 0.13), sex ($\beta = 0.06$; 95%-CI: 0.2, 0.1), perceived behavioral control ($\beta = 0.09$; 95%-CI: 0.05, 0.14), extraversion ($\beta = 0.05$; 95%-CI: 0.01, 0.09), attitude towards students with ADHD ($\beta = 0.06$; 95%-CI: 0.02, 0.11), and teachers affiliated with a primary or special needs school ($\beta = 0.04$; 95%-CI: 0.003, 0.09).

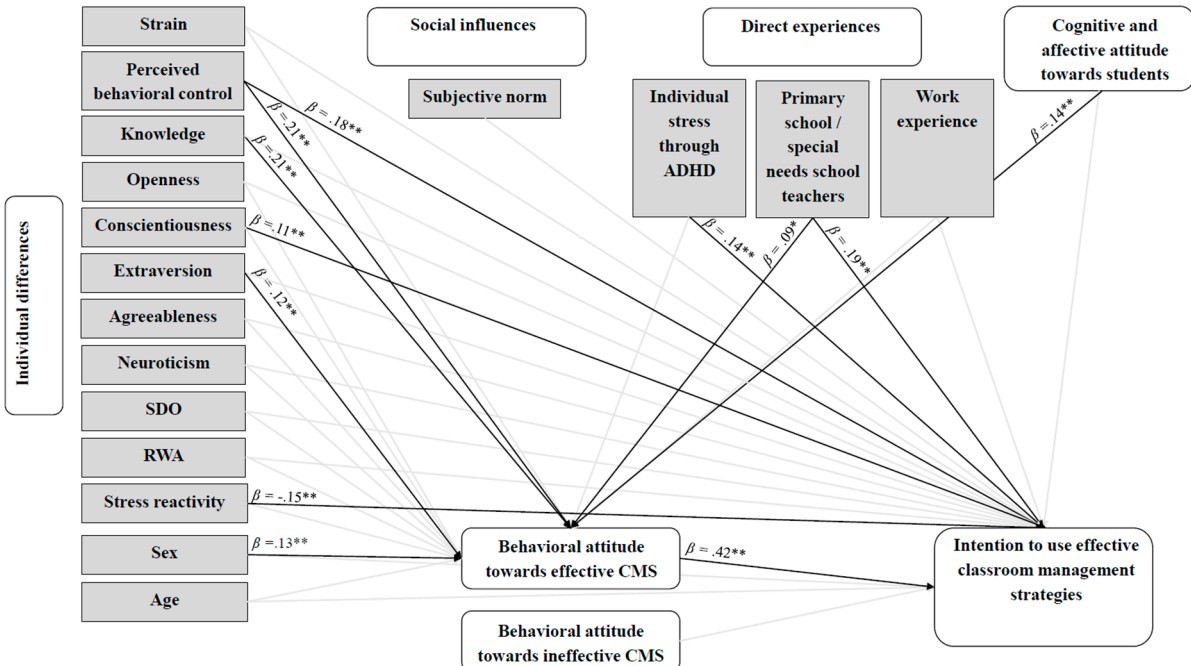

**Figure 2.** Path model showing the variables that influence the application effective classroom management strategies (CMS). Bold arrows represent significant relevant variables (* $p < 0.05$, ** $p < 0.01$). Path models' fit indices: $R^2_{\text{intention to use effective CMS}} = 0.47**$; $R^2_{\text{behav. attitude}} = 0.24**$, test of model fit: $\chi^2(2) = 1.13$, $p = 0.57$; test for the baseline model: $\chi^2(1) = 458.81$, $p = < 0.01$.; Comparative-Fit-Index (*CFI*) = 1.00; Tucker–Lewis-Index (*TLI*) = 1.00; *RMSEA* = 0.00; Standardized-Root-Mean-Square-Residual (*SRMR*) = 0.01.

### 3.2.2. Ineffective Classroom Management Strategies

Even if the direct replication of the model to explain the intended implementation of ineffective CMS was satisfactory, we calculated another model. Our aim was to find out whether additional variables would clarify a further intention's variance. In Figure 3, our analysis focusing on the intention to apply ineffective CMS is shown. Fit indices suggest a good model fit for the model regarding the intention to apply ineffective CMS with the added variables (test of model fit: $\chi^2(7) = 2.16$, $p = 0.95$; test for the baseline model: $\chi^2(39) = 300.89$, $p < 0.01$; *CFI* = 1.00; *TLI* = 1.00; *RMSEA* = 0.00; *SRMR* = 0.01) and a high variance explanation ($R^2_{\text{intention}} = 0.39$, $p < 0.01$; $R^2_{\text{behav. attitude}} = 0.10$, $p < 0.01$).

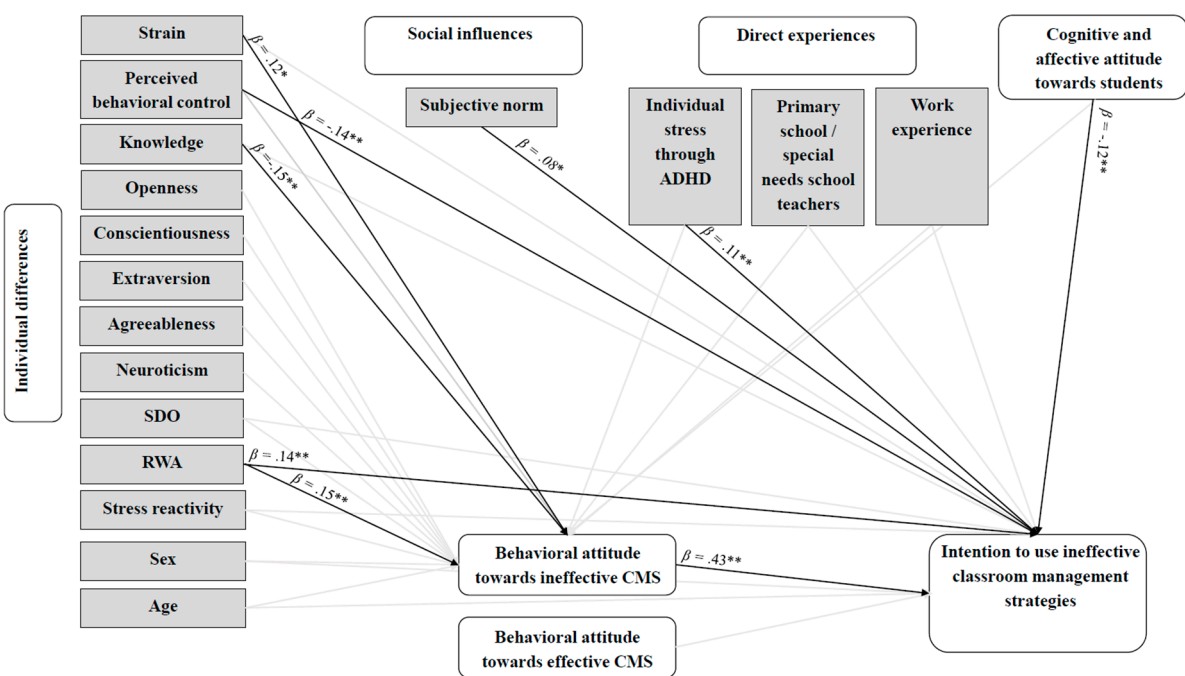

**Figure 3.** Path model showing the variables that influence implementation of ineffective CMS. Bold arrows represent significant relevant variables (* $p < 0.05$, ** $p < 0.01$). Path models' fit indices: $R^2_{\text{intention to use ineffective CMS}} = 0.39$**; $R^2_{\text{behav. attitude}} = 0.10$**; test of model fit: $\chi^2(7) = 2.16$, $p = 0.95$; test for the baseline model: $\chi^2(39) = 300.89$, $p = <.01.$; $CFI = 1.00$; $TLI = 1.00$; Root-Mean-Square-Error-of-Approximation ($RMSEA$) = 0.00; $SRMR = 0.01$.

Variance clarification regarding the intention to apply ineffective CMS is at 39%. Variables that contributed positively to variance explanation were behavioral attitude towards ineffective interventions ($\beta = 0.43$, $p < 0.01$), subjective norm ($\beta = 0.08$, $p < 0.05$), individual stress through ADHD ($\beta = 0.11$, $p < 0.01$), and RWA ($\beta = 0.14$, $p < 0.01$). Variables having a negative impact were perceived behavioral control ($\beta = -0.14$, $p < 0.01$) and attitude towards students with ADHD ($\beta = -0.12$, $p < 0.01$).

The amount of variance explained in the behavioral attitude towards ineffective interventions was 10% ($R^2 = 0.10$), implying low-to-moderate variance clarification [45]. Positively contributing variables were RWA ($\beta = 0.14$, $p < 0.01$), and strain ($\beta = 0.12$, $p < 0.05$). Knowledge was a negatively contributing variable ($\beta = -0.15$, $p < 0.01$).

All mediations with the intention to apply CMS as the dependent variable, and the behavioral attitude towards ineffective CMS were significant: RWA ($\beta = 0.06$; 95%-CI: 0.02, 0.11), strain ($\beta = 0.05$; 95%-CI: 0.01, 0.1), knowledge ($\beta = -0.07$; 95%-CI: −0.11, −0.2). Figure 3 shows all inter-correlations between variables.

In Table 2, all inter-correlations between the variables are listed.

**Table 2.** Inter-correlation between the investigated variables.

| | 1 | 2 | 3 | 4 | 5 | 6 | 7 | 8 | 9 | 10 | 11 | 12 | 13 | 14 | 15 | 16 |
|---|---|---|---|---|---|---|---|---|---|---|---|---|---|---|---|---|
| 1. Knowledge | 1 | - | - | - | - | - | - | - | - | - | - | - | - | - | - | - |
| 2. Age | 0.23 ** | 1 | - | - | - | - | - | - | - | - | - | - | - | - | - | - |
| 3. Work experience | 0.20 ** | 0.87 ** | 1 | - | - | - | - | - | - | - | - | - | - | - | - | - |
| 4. ADHD [1] experience | 0.03 | 0.31 ** | 0.32 ** | 1 | - | - | - | - | - | - | - | - | - | - | - | - |
| 5. Sex | 0.15 ** | −0.01 | 0.05 | −0.09 * | 1 | - | - | - | - | - | - | - | - | - | - | - |
| 6.Primary/special needs school teachers | 0.14 ** | 0.14 ** | 0.17 ** | −0.14 ** | 0.21 ** | 1 | - | - | - | - | - | - | - | - | - | - |
| 7. Attitude towards effective CMS | 0.29 ** | 0.14 ** | 0.15 ** | 0.02 | 0.20 ** | 0.19 ** | 1 | - | - | - | - | - | - | - | - | - |
| 8. Attitude towards ineffective CMS | −0.17 ** | −.010 * | −0.04 | 0.01 | −0.04 | −0.05 | −0.09 * | 1 | - | - | - | - | - | - | - | - |
| 9. Attitude towards students | −0.12 ** | −0.07 | −0.10 * | 0.01 | −0.07 | −0.04 | 0.17 ** | −0.02 | 1 | - | - | - | - | - | - | - |
| 10. Perceived control | 0.10 * | 0.12 ** | 0.17 ** | 0.13 ** | 0.00 | 0.13 ** | 0.33 ** | 0.01 | 0.32 ** | 1 | - | - | - | - | - | - |
| 11. Subjective norm (shortened) | −0.04 | −0.08 | −0.09 * | −0.07 | 0.01 | 0.07 | −0.03 | −0.03 | −0.07 | −0.16 ** | 1 | - | - | - | - | - |
| 12. PSRS | −0.04 | −0.10 * | −0.10 * | −0.14 ** | 0.21 ** | 0.08 | −0.11 * | 0.05 | −0.20 ** | −0.23 ** | 0.35 ** | 1 | - | - | - | - |
| 13. Strain (BSI) | −0.09 * | −0.07 | −0.07 | 0.04 | −0.06 | 0.03 | −0.10 * | 0.14 ** | −0.04 | −0.11 ** | 0.01 | 0.31 ** | 1 | - | - | - |
| 14. Stress (self - report) | 0.07 | 0.06 | 0.06 | 0.00 | 0.15 ** | 0.20 ** | −0.08 | 0.02 | −0.37 ** | −0.31 ** | 0.19 ** | 0.26 ** | 0.09 * | 1 | - | - |
| 15. Application of effective CMS | 0.26 ** | 0.20 ** | 0.24 ** | 0.06 | 0.24 ** | 0.35 ** | 0.55 ** | 0.03 | 0.01 | 0.32 ** | −0.05 | −0.11 ** | −0.09 * | 0.13 ** | 1 | - |
| 16. Application of ineffective CMS | −0.13 ** | −0.15 ** | −0.11 ** | −0.04 | 0.02 | −0.03 | −0.21 ** | 0.46 ** | −0.24 ** | −0.27 ** | 0.17 ** | 0.22 ** | 0.11 * | 0.24 ** | −0.08 | 1 |
| SDO | −0.18 ** | −0.11 ** | −0.05 | −0.01 | −0.10 * | −0.05 | −0.14 ** | 0.16 ** | −0.13 ** | −0.08 * | −0.07 | −0.02 | 0.02 | 0.06 | −0.04 | 0.15 ** |
| RWA | −0.13 ** | −0.07 | −0.05 | −0.03 | 0.02 | 0.05 | −0.09 * | 0.18 ** | −0.16 ** | 0.12 ** | 0.07 | 0.13 ** | 0.04 | 0.18 ** | 0.04 | 0.24 ** |
| Openness | 0.07 | 0.09 * | 0.05 | 0.07 | 0.03 | −0.01 | 0.09 * | 0.03 | 0.05 | 0.19 ** | −0.04 | −0.07 | 0.00 | 0.00 | 0.14 ** | −0.07 |
| Conscientiousness | 0.14 * | 0.08 | 0.08 * | 0.06 | 0.17 ** | 0.04 | 0.15 ** | −0.03 | −0.04 | 0.12 ** | −0.04 | −0.09 * | −0.17 ** | 0.00 | 0.24 ** | −0.09 * |
| Extraversion | 0.07 | 0.01 | 0.01 | 0.03 | 0.05 | −0.02 | 0.17 ** | 0.04 | 0.04 | 0.16 ** | −0.12 ** | −0.24 ** | −0.07 | −0.04 | 0.15 ** | −0.09 * |
| Agreeableness | 0.13 ** | 0.18 ** | 0.15 ** | −0.02 | 0.09 * | 0.05 | 0.20 ** | −0.09 * | 0.08 | 0.16 ** | 0.03 | −0.10 * | −0.10 * | −0.02 | 0.14 ** | −0.16 ** |
| Neuroticism | −0.09 * | −0.12 ** | −0.12 ** | −0.13 ** | 0.13 * | 0.06 | −0.15 ** | 0.05 | −0.18 ** | −0.26 ** | 0.22 ** | 0.64 ** | 0.38 ** | 0.24 ** | −0.11 ** | 0.20 ** |

Notes: [1] Attention deficit/hyperactivity disorder (ADHD); ** $p < 0.01$; * $p < 0.05$.

## 4. Discussion

In the present study we aimed to examine whether the facilitators and barriers regarding the implementation of CMS supporting students with ADHD that are important for pre-service teachers are replicable among in-service teachers. We additionally analyzed whether extending our model with further variables of potential relevance when examining in-service teachers would enhance the clarification of variance.

### 4.1. First Research Question ("Can the Model to Apply (In-) Effective CMS for Pre-Service Teachers Be Replicated in a Sample of In-Service Teachers?")

The model that worked best to explain variance in the intention to apply ineffective CMS for pre-service teachers results in a good fit for in-service teachers (36% clarified variance in the intention to use ineffective CMS). Primarily, the behavioral attitude towards ineffective CMS contributed significantly to variance explanation, whereas the model aiming to explain the intention to apply effective CMS was not satisfactory.

The major disadvantage of this direct replication is that variables describing differences between in-service teachers like work experience were not included as they could not be assessed with our sample of pre-service teachers as they were so homogeneous. Most likely, this factor clarifies why our model explaining the intention to apply effective CMS revealed an unsatisfactory model fit. Therefore, our next research question aimed to determine whether extending our model with variables that do justice to that fact would improve the intentions variance clarification.

### 4.2. Second Research Question ("Does Expanding the Model for In-Service Teachers Result in an Improved Model Fit?")

Relying on the aforementioned results, we conducted two further path analyses that included more predictors. Due to differences between pre- and in-service teachers, we investigated whether variables that do justice to those (sex, age, work experience assessed by years on the job, and type of school) would enable us to deepen our knowledge about the intention to apply (in-)effective CMS. It follows that replicating with those variables led in both cases to a better model fit, implying that the added variables are important for variance clarification, especially regarding effective CMS. The extended model clarifies 47% of the intention to apply effective CMS and 39% of the intention to apply ineffective CMS. The following sections refer to all variables included in the extended models.

### 4.3. Extended Model Regarding Effective CMS

In comparison to the model that fits best for pre-service teachers, the added variables make a strong explanatory contribution. Furthermore, every variable influencing the behavioral attitude towards effective CMS also influences the intention to apply effective CMS mediated through behavioral attitude.

#### 4.3.1. Attitude towards Effective CMS and Students with ADHD

The behavioral attitude towards effective CMS is the most important variable regarding variance clarification towards the intention to apply effective CMS. This is consistent with the finding in pre-service teachers [20] and an Austrian study including pre-service and in-service teachers [22]. In concordance with Ajzen [21], behavioral attitude reveals a stronger impact than the attitude towards students with ADHD. The latter is important for the variance clarification towards the behavioral attitude. However, it does not clarify variance in the intention to apply effective CMS, which is in line with the findings in pre-service teachers and the Austrian sample [20,22]. Furthermore, it supports the assumption that a more positive attitude towards students with ADHD strengthens the intention to apply effective CMS indirectly mediated through the behavioral attitude, which is in line with the TPB [21].

Dort et al. [47] identified three different latent classes of teachers regarding their attitude towards students with ADHD. The attitude-profile class assumed to be the most

helpful for students with ADHD is, surprisingly, not associated with the best attitude score. Having a realistic attitude that differentiates the positive aspects of students as positive and those negative as negative is more often associated with implementing effective CMS, although the most positive attitude on average was not associated with any particular application of effective or ineffective CMS. This fact may explain why a pooled measurement of attitude towards students with ADHD makes an only conditional contribution.

### 4.3.2. Direct Experiences

Individual stress through ADHD revealed a positive contribution towards the intention to apply effective CMS. Individual stress is not relevant to pre-service teachers in conjunction with their intention to apply effective CMS [20]. Nevertheless, a certain amount of stress triggered by children with ADHD seems to be beneficial for in-service teachers in helping them deal positively with the classroom challenges that children with ADHD induce. This is in line with research on stress demonstrating that performance under pressure initially increases [48].

The teachers' affiliation with primary or special needs schools showed likewise a positive contribution towards the intention to apply effective CMS. Primary and special needs school teachers might undergo different training than teachers from other types of school. This training might focus more on managing strenuous classroom behavior. Furthermore, there might be a difference in attitude, personality and knowledge between teachers who choose to be special needs or primary school teachers compared to teachers of other schools. These two aspects might explain why teachers' affiliation with primary or special needs school reveals a positive association with the behavioral attitude towards effective CMS and with the intention to apply effective CMS. Thus the training that teachers at other school types undergo might benefit from absorbing aspects from the training offered to student teachers planning to work in a special needs or primary school. Work experience measured by years on the job showed no relevant contribution.

### 4.3.3. Social Influences

Subjective norm revealed no relevant impact on the intention to apply CMS or the behavioral attitude towards effective CMS, a finding in line with the evidence from pre-service teachers, the Austrian study sample, and other investigations in the school context [20,22,26].

### 4.3.4. Individual Differences

Conscientiousness and extraversion are two personality variables demonstrating positive effects. They might therefore be beneficial for creating a positive attitudes towards CMS and their implementation [49]. In contrast to the models with pre-service teachers and unlike in the Austrian study sample, we found that psychological strain made no relevant contribution to the intention or attitude towards effective CMS among in-service teachers [20,22]. However, stress reactivity, which characterizes the troublesome handling of stressful situations [33], revealed a negative impact on the intention to apply effective CMS, highlighting that if teachers are better at handling stressful situations, they may also be more likely to apply or feel more strongly the intention to implement effective CMS.

Knowledge makes a positive contribution towards the behavioral attitude and therefore supports the assumption that basic knowledge is important, but not sufficient to enhance the intention to apply effective CMS [20,50]. Perceived behavioral control is relevant both for the behavioral attitude towards effective CMS and for the intention to apply effective CMS; however, we did not observe this effect among pre-service teachers [20]. The expectation that one is capable of applying effective CMS is one of the most important variables in the present study. This reinforces findings regarding perceived behavioral control: In the context of implementing supportive strategies for students in an inclusive context, perceived behavioral control is similarly relevant [26].

Furthermore, there is a sex difference in CMS implementation: Female sex contributes positively to the behavioral attitude towards effective CMS, in line with the (rare) research on sex differences in teaching. For example, a study by Sternglanz and Lyberger-Ficek [35] showed that male teachers prefer teaching male students, whereas no difference was observed in female-headed classrooms. This finding concurs with the present study's, namely that in contrast to their male colleagues, women do not seem to favor any special group of students. However, the aforementioned study is quite dated, and sex differences in teaching have received little attention since then.

### 4.4. Extended Model Regarding Ineffective CMS

Behavioral attitude as well as the attitude towards students with ADHD, individual differences, direct experiences, and social influences all contribute to explaining variance. Furthermore, every variable that makes a contribution towards the behavioral attitude towards ineffective CMS also has an influence on the intention to apply ineffective CMS, mediated through behavioral attitude.

#### 4.4.1. Attitude towards Ineffective CMS and towards Students with ADHD

Behavioral attitude towards ineffective CMS is the most important variable regarding variance clarification in the intention to apply ineffective CMS, supporting the Austrian study's conclusion and the study with pre-service teachers that assumed that the strategy applied with the desired effect is the most important variable [20,22]. Attitude towards students with ADHD makes a negative contribution towards the intention to apply ineffective CMS, which differs from the pre-service teachers' finding [20], but is in line with the Austrian study results [22]. This finding implies that a more positive attitude towards students with ADHD might weaken the intention to apply ineffective CMS.

#### 4.4.2. Direct Experiences

Individual stress through ADHD makes a positive contribution towards the intention to apply ineffective CMS. This is interesting, as it also makes a positive contribution towards the intention to apply effective CMS. It might be that participants who feel more stressed by students with ADHD feel a stronger intention to apply any kind of CMS, whether they are effective or not. This explanation might be supported by teachers' lack of knowledge especially when it comes to ADHD treatment [51]. It so happens that teachers want to react to stressful students, but together with their little knowledge they then strongly intend to apply any kind of CMS.

This finding is of particular importance, as Westman and Etzion [52] showed that there is a crossover effect regarding school stress between teachers and their principals, implying that every stressed teacher contributes to creating a tense mood in the school, reinforcing a generally bad mood and vicious circle. The variables work experience assessed by years on the job and the affiliation with primary and special needs schools did not contribute significantly to variance clarification.

#### 4.4.3. Social Influences

The subjective norm contributes to the intention to apply ineffective CMS. This finding differs from those from pre-service teachers and the Austrian study [20,22] and implies that a higher rating on subjective norm might trigger a stronger intention to apply ineffective CMS. This in turn might indicate that implementing ineffective CMS is a more socially recognized behavior and therefore this kind of CMS is applied more consistently by in-service teachers [16]. Another potential explanation is that teachers want to do something but are unable to differentiate between effective and ineffective CMS. Other research focusing on (changing) teacher behavior has also suggested that subjective norm is relevant and that implementing different strategies in the classroom depends primarily on other teachers, headmasters, and parents [53,54]. To convince teachers to avoid implementing ineffective CMS, a general rethink must take place.

### 4.4.4. Individual Differences

RWA exhibited a positive relationship with the behavioral attitude towards and intention to apply ineffective CMS—which makes sense, as it represents submission to authority, strict adherence to conventional standards, and general authority-sanctioned aggressivity toward others [55]. This suggests that teachers stick to applying ineffective CMS, as those are more traditional and emphasize the hierarchy between students and teachers. According to Lottie [56], more conservative, authoritarian teachers are more resistant to change because they are pursuing a system-immanent career and have developed their beliefs about teaching and learning mainly from their own, mostly positive experiences when they were in school [57]. Ineffective CMS often have positive short-term effects, e.g., getting the student to leave the classroom without comment leads immediately to less interference. However, these strategies do not reduce strenuous, annoying behavior over the long term, as they tend to reinforce it because the misbehaving student has no chance to learn alternative behavior (even though teachers may have had short-term positive experiences with them).

Psychological strain exhibited a positive relationship with the behavioral attitude towards ineffective CMS, thus supporting the assumption that greater strain in teachers leads to less engagement and therefore more frequent application of ineffective CMS in the classroom and a worse teacher-student relationship [58,59].

Knowledge revealed a negative impact on the variance clarification of the behavioral attitude, but not on the intention to apply ineffective CMS. Ajzen and colleagues [50] claimed that knowledge is necessary, but not enough for the intention to do something directly—in line with our finding. Nevertheless, there is a knowledge gap in teachers regarding the handling of ADHD in the classroom, and it helps to differentiate between effective and ineffective CMS. We need to close this knowledge gap [51].

Perceived behavioral control demonstrated a negative relationship with the intention to apply ineffective CMS—exactly the same finding as in our sample of pre-service teachers and in the Austrian study [20,22].

### 4.5. Implications

The behavioral attitude, and therefore expectation that a specific CMS will prove effective is the most important variable regarding the intention to apply a specific CMS in pre- and in-service teachers. The attitude towards students with ADHD proved to be important indirectly [20,22]. Those findings highlight the need for experimental testing to change attitudes about and therefore expectations towards CMS and towards students with ADHD.

A preliminary investigation by Barnett, Corkum and Elik [60] showed that a web-based intervention changed teachers' attitudes and self-reported competence in teaching, an indication of what might be beneficial. We are planning a virtual reality investigation to examine how attitude-related expectations can be changed. The expectations about CMS and therefore attitude towards those CMS are relevant, and research should thus focus on changing those expectations. An experiment could be set up in which the reaction of the students to certain CMS differs from their teachers' expectations. One could then examine if and how teachers' expectations change.

Direct experiences, such as individual stress through ADHD are important in clarifying variance in the intention to apply effective and ineffective CMS and in the behavioral attitude towards effective CMS. Low-intensity stress from children with ADHD seem to be beneficial in conjunction with the intention to implement effective CMS, while more severe stress increases the likelihood to apply ineffective CMS. The handling of students perceived to be hard to manage and strenuous is therefore an important topic to address more intensively; it would be an interesting approach to investigate whether the stronger implementation of effective CMS reduces such stress and if that in turn leads stronger intention to apply effective CMS.

Another important direct experience in clarifying variance in the intention to apply effective and ineffective CMS and in the behavioral attitude towards effective CMS is teachers' affiliation with primary and special needs school (but not work experience as years doing the job). Teachers at primary and special needs schools have a more positive attitude towards effective CMS and show a stronger intention to implement them. On the one hand, this can be because there is a self-selecting process in advance, i.e., which type of school was chosen, but on the other hand, those teachers might undergo different training, e.g., regarding ADHD. Their education concerning ADHD could set a good example for the training of teachers at other types of school.

Social influences like the subjective norm show a positive relationship with the intention to apply ineffective CMS, suggesting that teachers are assuming that ineffective CMS are appropriate. For attempts to change attitudes and for the intention to apply CMS to succeed, this fact needs to be considered. Overall, effective CMS must become well established and popular among teachers.

Individual differences, like conscientiousness, extraversion, RWA, strain, stress reactivity, knowledge, perceived behavioral control, and sex show a diverging amount of variance clarification. Conscientiousness, extraversion, and RWA are three person-inherent variables that make it easier or harder for teachers to integrate effective CMS and reject ineffective ones. Psychological strain is positively associated with ineffective CMS, and stress reactivity negatively with effective CMS. A Germany-wide investigation showed that primary teachers suffer from work-related psychosocial exhaustion and that the reasons for this are physical problems and an effort-reward imbalance [61,62]. This suggests that handling stress and improving mental health are key influencing factors to target. One approach would be to integrate a relaxation program [63], which could be the first step in reducing the pressure on teachers. We need to examine whether that would in turn lead to the better implementation of CMS. Both knowledge and perceived behavioral control exert a preventive effect, as they reinforce directly and indirectly the intention to apply effective CMS and dampen the intention to apply ineffective CMS. For example, it would be worthwhile to develop further the aforementioned web-based intervention by Barnett and colleagues [60] in this regard.

With regard to knowledge, it can be seen that this is an important influencing factor for both pre- and in-service teachers, promoting the use of effective strategies and reducing those that are less effective. This should imply that not only more knowledge about and how to deal with ADHD should be imparted during the course of studies, but also that practicing teachers should get more knowledge in order to improve dealing with students with ADHD. Furthermore, it needs experimental or quasi-experimental testing with observational studies to investigate how and if teachers implement CMS after they got the knowledge about ADHD and CMS. With pre-service teachers, a special focus could be placed on the fact that, after their internship, they sum up the extent to which they have used CMS and with what success.

### 4.6. Limitations

The present study is based on an online questionnaire; thus, our results depend on the participants' own-assessments and could thus be biased. This distortion can be exacerbated and caused by the fact that the initial drop-out rate is over 70%. However, this is the case with many online surveys, as many click them, but then do not edit the survey any further Furthermore, we were unable to investigate whether the group of teachers who dropped out differed from those who completed the survey. Moreover, we can make no statements about causal influences as our results, are of correlative nature due to the cross-sectional design. There is a female bias in our study as we included more female teachers than males. This might be due to the self-selecting nature of online surveys, but in general there are more female than male teachers in the school context.

As mentioned in our first investigation [20], the literature reveals an intention-behavior gap [64] indicating that intention and behavior are not closely related, and major change

in the intention only leads to a small-to-medium change in the belonging behavior [64]. Therefore, we can only make assumptions but no clear predictions about teachers' implementation of CMS. Furthermore, in line with Ajzen [21,50,65], retrospective contemplation of one's own behavior is often biased, and contrary to the intention, a poor predictor of subsequent behavior. In summary, it would be preferable to assess the behavior of interest specifically, which would necessitate an experimental investigation or natural setting.

## 5. Conclusions

Implementing CMS in the classroom leads to a lasting reduction of ADHD symptoms. CMS can be used with other students and results in a sustainable conservation of resources by reducing burden on teachers and preventing the recurrence of ADHD-symptoms. Behavioral attitude and therefore the expectation that a specific CMS is (in)effective is the most important variable regarding variance clarification in the intention to apply CMS in pre- and in-service teachers in Germany and Austria. The attitude towards students with ADHD, as well as personal experiences, social influences, and individual differences are important as well. Generally speaking, ADHD-specific knowledge and perceived behavioral control might be easy to influence and demonstrate the desired effect of eventually enhancing the intention to apply effective CMS. Further, experimental investigations and training interventions for pre- service and in-service teachers are needed to finally close this gap between the scientific evidence and actual practice.

**Author Contributions:** Conceptualization: A.E.S., M.D., and H.C.; methodology: A.E.S., M.D., M.S., and H.C; validation: M.D., M.S., and H.C.; formal analysis: A.E.S.; investigation: A.E.S. and M.D.; resources: H.C.; data curation: A.E.S., M.D., and H.C.; writing—original draft preparation: A.E.S.; writing—review and editing: H.C., M.S., and M.D.; visualization: A.E.S.; supervision: H.C. and M.S.; project administration: H.C. and M.S.; funding acquisition: H.C. and M.S. All authors have read and agreed to the published version of the manuscript.

**Funding:** This research was part of the project "ADHD in the classroom" that is part of the RTG 2271 and funded by the German Research Foundation (DFG) project number 290878970-GRK 2271, project 1.

**Institutional Review Board Statement:** Ethical review and approval were waived for this study, due to specifications of the University of Marburg, since no vulnerable samples are examined, and the study does not violate the physical or psychological integrity, the right to privacy or other subjective rights or the interests of study participants.

**Informed Consent Statement:** Informed consent was obtained from all subjects involved in the study.

**Data Availability Statement:** The datasets are stored at the University of Marburg and can be accessed there.

**Conflicts of Interest:** The authors declare no conflict of interest.

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
