# Peer review of "Influences on Teachers’ Intention to Apply Classroom Management Strategies for Students with ADHD: A Model Analysis"

_sustainability, doi:10.3390/su13052558_

Round 1
Reviewer 1 Report
The article proposes a model that tries to explain the variables that determine the use of classroom management strategies for students with ADHD by their teachers.
For this, data from 599 in-service teachers are used.
Previously, the authors of the article themselves had published results of a similar model using data from 1.086 pre-service teachers (this study is cited as reference 20).
The article is correctly written and organized, the analysis are rigorous and the subject is of interest to potential readers.
My general assessment is that this is a good article that requires some minor changes that I discuss below.
1. Lines 153-155.
"As pre-service teachers (unlike those in in service) might differ in knowledge, this study aims to clarify potential similarities and differences between groups of pre- and in-service teachers [20,34]".
This sentence is somewhat ambiguous.
"This study" refers to the study of this article?
In this case, these lines should be relocated to section "1.4 The present study".
If, on the contrary, it refers to study 34, it should be clarified.
2. Lines 192-193.
"The survey was started by N = 2320 potential participants".
The number of 2320 potential participants is substantially different from the 599 teachers who actually participated in the study.
This issue should be explained and justified.
As expressed, it is confusing information.
3. Does the study have the consent of an ethics committee?
This question should be stated in the method section or in the final statements of the article.
4. Lines 289-293.
If the instrument has previously demonstrated good psychometric properties in the same study context (as appears to be the case according to reference 38), it is not necessary to provide this information in the article.
Likewise, in the rest of the instruments used in the study that have already shown good psychometric data, it is not necessary to offer again Cronbach's alpha values.
5. Lines 322-325.
This information appears previously, in section 2.1. It should not be repeated here.
6. Subjective norms instrument.
It should make clarified that the version that was finally used was the shortened one (maybe after line 284).
7. Lines 492-493.
"Furthermore, there might be a difference between teachers who choose to be special needs or primary school teachers compared to others".
7.1.Review the construction of the sentence.
7.2. It should be specified what these differences are (probably related to motivation or attitudes themselves).
8. Implications.
In the implications of the study, some possible implications in the initial and continuing training of teachers should be discussed, both from a theoretical point of view (acquisition of knowledge) and in the teaching experience with children with ADHD (especially in the school practice periods of pre-service teachers).
9. Formal issues:
9.1. Keywords must be in alphabetic order.
9.2. Line 485. There is a missprint. It should be written "teachers in" instead of "teacher sin".
9.3. Table 2. The margins of table 2 do not allow to see the name of the last variables.
Author Response
Please see the attachement.

Reviewer 2 Report
The indicated sample has a female bias (17.7% males, and 82.3% females), justifying the non-existence of this.
The first two points and 4.3. The discussion section does not present a real "discussion." and sections 4.3.2 and 4.4. should be improved in that sense.
Reviewer 3 Report
Review of Article No: sustainability-1097604
Title: Influences on teachers’ intention to apply classroom management strategies for students with ADHD: A model analysis
Authors: Anna Enrica Strelow, Martina Dort, Malte Schwinger and Hanna Christiansen
Journal: Sustainability (ISSN 2071-1050)
Contribution of the Article
In their article, the authors present a model analysis used to study influences on teachers’ intention to apply classroom management strategies for students with attention-deficit/hyperactivity disorder. The article represents thorough and original research in a topical area.
The title explicitly reflects the contents of the paper.
The abstract presents a clear and objective picture of the article by summarizing the main points of the text.
In the introduction, the authors outline the ADHD condition and the related symptoms students exhibit in class. Emphasis is placed on evidence-based classroom management strategies and their effectiveness in reducing unfavourable classroom behaviour. The reasons for the gap between science and practice are sought in order to explain why teachers rely more often on ineffective CMS, such as threatening a student with ADHD with punishment, than effective ones. A number of studies are referred to to demonstrate that the behavioural attitude towards CMS is the most important variable in explaining variance in the intention to apply effective and ineffective CMS. Further variables are defined, which in turn influence teachers’ behavioural attitude.
Influences on teachers’ attitude and intention to apply CMS are considered, and more precisely direct experiences with students affected by ADHD (work experience), social influences, namely the willingness to behave according to the expectations of relevant peers, and individual differences.
Based on the outlined differences in some variables between pre-service and in-service teachers, the authors set out to find if the model for pre-service teachers can be repeated in a sample of in-service teachers and whether expanding upon the model with the variables sex, age, work experience, and type of school improves the model fit, so they pose the following two questions:
- Can the model to implement (in-)effective CMS for pre-service teachers (with the variables attitudes, direct experiences, social influences, individual differences) be replicated in a sample of in-service teachers?
- Does expanding the model for in-service teachers by considering the variables sex, age, work experience according to years on the job and type of school result in an improved model fit?
In the second section, Materials and Methods, the survey is delineated regarding the participants and the procedure and sequence of the survey. The ADHD School Expectation Questionnaire is used to assess CMS of 237 teachers when handling ADHD-related behaviour in their classrooms and scales of the values for Cronbach’s α, means and standard deviations are defined. Attitudes towards students with ADHD and towards classroom management strategies are assessed as well as direct experiences, social influences, individual differences, knowledge, perceived behavioural control, and sex and age. IBM® SPSS®24.0 and Mplus 8.4 are used to conduct the statistical analyses.
In the third section, the authors present the results of their research by answering the questions that they posed earlier. To answer the second question, the authors conduct two further analyses of effective and ineffective classroom management strategies with the variables sex, age, primary school/special needs school teachers, work experience assessed in years, and school type that teachers are educated to teach at.
In the fourth section, Discussion, the authors present in detail the findings in their study, which aims to examine whether the facilitators and barriers regarding the implementation of CMS supporting students with ADHD that are important for pre-service teachers are replicable among in-service teachers and whether extending their model with further variables of potential relevance when examining in-service teachers would enhance the clarification of variance. Additionally, the authors consider the limitations of their research results stating that they are assumptions and not clear predictions about teachers’ implementation of CMS and that it would be preferable to assess the behaviour of interest specifically, which would necessitate an experimental investigation or natural setting.
Finally, the authors draw conclusions based on their findings and point out that experimental investigations and training interventions for pre-service and in-service teachers are needed to finally bridge the gap between the scientific evidence and actual practice.
The references provide a sufficient number of scientific materials for the reader who is interested in exploring further the subject matter of the article.
Language of the article
The article is written in fluent English. There are just a few very minor inaccuracies such as the following:
“This expectations and therefore behavioral attitude are influenced ..” (these) (lines 90-91)
“In CMS, terms, this ..” (the first comma is redundant) (line 163)
“Teachers individual stress levels ..” (lack of apostrophe) (266)
implementaion (666)
own’s own behavior (667)
Conclusion
In conclusion, I can say that the article is well structured, scientifically sound, accurate, and complete. It is informative and interesting to read and it expands and enhances the ongoing research in its subject field. The authors have followed the guidelines of the journal.
